# Bottleneck or Crossroad? Problems of Legal Sources Annotation and Some Theoretical Thoughts

**Amedeo Santosuosso ***[ID] **and Giulia Pinotti**

Research Center ECLT, University of Pavia, 27100 Pavia PV, Italy; giulia.pinotti01@ateneopv.it
* Correspondence: a.santosuosso@unipv.it

**Abstract:** So far, in the application of legal analytics to legal sources, the substantive legal knowledge employed by computational models has had to be extracted manually from legal sources. This is the bottleneck, described in the literature. The paper is an exploration of this obstacle, with a focus on quantitative legal prediction. The authors review the most important studies about quantitative legal prediction published in recent years and systematize the issue by dividing them in text-based approaches, metadata-based approaches, and mixed approaches to prediction. Then, they focus on the main theoretical issues, such as the relationship between legal prediction and certainty of law, isomorphism, the interaction between textual sources, information, representation, and models. The metaphor of a crossroad shows a descriptive utility both for the aspects inside the bottleneck and, surprisingly, for the wider scenario. In order to have an impact on the legal profession, the test bench for legal quantitative prediction is the analysis of case law from the lower courts. Finally, the authors outline a possible development in the Artificial Intelligence (henceforth AI) applied to ordinary judicial activity, in general and especially in Italy, stressing the opportunity the huge amount of data accumulated before lower courts in the online trials offers.

**Keywords:** legal sources; prediction; legal analytics

---

## 1. Legal Analytics and Quantitative Legal Prediction

In his leading work on artificial intelligence and legal analytics, Kevin Ashley sharply states that "while AI and law researchers have made great strides, a knowledge representation bottleneck has impeded their progress toward contributing to legal practice" [1] (p. 29). The bottleneck is described in very clear terms: "So far, the substantive legal knowledge employed by their computational models has had to be extracted manually from legal sources, that is, from the cases, statutes, regulations, contracts, and other texts that legal professionals actually use. That is, human experts have had to read the legal texts and represent relevant parts of their content in a form the computational models could use." (p. 29).

This paper is an exploration of this obstacle and an overview of its many components and aspects, with a focus on quantitative legal prediction. In a sense, our contribution aims, among many others, at opening the bottleneck. Indeed, the discovery of knowledge that can be found in text archives is also the discovery of the many assumptions and fundamental ideas embedded in law and intertwined with legal theories. Those assumptions and ideas can be seen as roads crossing within the bottleneck. Their analysis can clarify some critical aspects of the bottleneck in quantitative legal prediction. In Section 2, we list the main technical and theoretical problems, which appear to be, at a first sight, the main components of the bottleneck. In Section 3, we review the most important studies about quantitative legal prediction published in recent years and try to systematize the issue by dividing them in text-based approaches, metadata-based approaches, and mixed approaches to prediction. In particular, we focus on the studies conducted by Aletras et al. (2016) [2] on the decisions

of the European Court of Human Rights (henceforth ECHR), Zhong et al. (2019) [3] on US Board of Veterans' Appeals, Katz et al. (2014) [4] on the Supreme Court decisions, and Nay (2017) [5] on the probability that any bill will become law in the U.S. Congress. Section 4 focuses on the main theoretical issues, such as the relationship between legal prediction and certainty of law, isomorphism, the interaction between textual sources, information, representation, and models, with a final sketch of the many features and facets of law today and their implications in legal prediction. Section 5 is dedicated to some applicative implications of the relationship between a text-based approach and the structure of a decision, and gives some information about some works in progress, with hints at possible, even institutional, developments. Finally, in Section 6, we show a possible path for future research in legal prediction and make a statement of support for an Italian initiative in this field.

## 2. Bottleneck and Crossroad: Two Metaphors for Text Analytics

Kevin Ashley [1], after describing the bottleneck, sees a light at end of the tunnel: "The text analytic techniques may open the knowledge acquisition bottleneck that has long hampered progress in fielding intelligent legal applications. Instead of relying solely on manual techniques to represent what legal texts mean in ways that programs can use, researchers can automate the knowledge representation process." [1] (p. 5). Text analytics "refers to the discovery of knowledge that can be found in text archives" and "describes a set of linguistic, statistical, and machine learning techniques (that model and structure the information content of textual sources for business intelligence, exploratory data analysis, research, or investigation". When the texts to be analyzed are legal, these techniques are called "legal text analytics or more simply legal analytics" [1,4].

Finally, Ashley makes a list of intriguing questions, such as "can computers reason with the legal information extracted from texts? Can they help users to pose and test legal hypotheses, make legal arguments, or predict outcomes of legal disputes?" and, at the end, replies as follows:

"The answers appear to be 'Yes!' but a considerable amount of research remains to be done before the new legal applications can demonstrate their full potential" [1] (p. 5).

The considerable amount of research Ashley refers to is something that looks like a bunch of roads with several crossroads, rather than a straight (even if long) path. In other words, a bottleneck and crossroads are two metaphors waiting to be unpacked (which is a metaphor as well!). Indeed, the knowledge we want to discover in text archives through legal analytics is all one with the many assumptions and fundamental ideas embedded in law and intertwined with legal theories. Additionally, each of these probably requires different approaches, which might allow us to "discern" or, at least, "not misunderstand" their existence and consistence.

Hereinafter, we just make some examples among the many, reserving Section 4 to a more in-depth discussion about them.

We can firstly consider differences in common law and civil law systems. It is clear that a rule-based approach can be envisaged at work only in the latter, these being the first a case-based system by definition. Additionally, we know that rule-based decision-making, at least in the pure meaning used by Frederick Schauer [6], is more easily represented in an if/then logic. Indeed, according to this author, a rule is not simply a legal provision, as it has to be written and clearly defined in its content: "rule-governed decision-making is a subset of legal decision-making rather than being congruent with it" (p. 11).

At a deeper level, the rationalistic/non-rationalistic divide is at the forefront and shows how the conceptual distance between code-based legal systems and the case law systems are shorter than usually supposed. In light of this, we may say that the enduring opposition between the rationalistic and the pragmatic (or historical) approaches to the law crosscuts even the civil law–common law fields and probably is the most important divide in contemporary law, in an era where common law and civil law systems have lost many of their original features.

In addition, moving toward the boundary between computation and law, it is worth noting that even in the AI field there is a divide between logicist and non-logicist approaches, which can

be partitioned into symbolic but non-logicist approaches, and connectionist/neurocomputational approaches. Even though some similarities might be tempting, establishing a direct correspondence between the two approaches in law and AI is unwarranted [7].

Going back to the legal field, some authors maintain that "we are fast approaching a third stage, 3.0, where the power of computational technology for communication, modeling, and execution permit a radical redesign, if not a full replacement, of the current system itself. [ . . . ] In the law, it turns out that computer code is considerably better than natural language as a means for expressing the logical structure of contracts, regulations, and statutes". In other words, Goodenough [8] is drawing a scenario where computer code supplants natural language in expressing the logical structure of typical manifestations of legal practice, such as contracts, regulations, and statutes.

Other authors, such as Henry Prakken, a professor at the Utrecht University (NL), and Giovanni Sartor, a professor at the European University Institute of Florence (I) [9], start from the point that "the law is a natural application field for artificial intelligence [and that . . . ] AI could be applied to the law in many ways (for example, natural-language processing to extract meaningful information from documents, data mining and machine learning to extract trends and patterns from large bodies of precedents)" [9] (p. 215). However, they stress that: "law is not just a conceptual or axiomatic system but has social objectives and social effects, which may require that a legal rule is overridden or changed. Moreover, legislators can never fully predict in which circumstances the law has to be applied, so legislation has to be formulated in general and abstract terms, such as 'duty of care', 'misuse of trade secrets' or 'intent', and qualified with general exception categories, such as 'self-defense', 'force majeure' or 'unreasonable'. Such concepts and exceptions must be interpreted in concrete cases, which create uncertainty and room for disagreement. This is reinforced by the fact that legal cases often involve conflicting interests of opposing parties. [ . . . ] All these aspects of the law, i.e., its orientation to future and not fully anticipated situations, the tension between the general terms of the law and the particulars of a case, and the adversarial nature of legal procedures, make that legal reasoning goes beyond the literal meaning of the legal rules and involves appeals to precedent, principle, policy and purpose, and involves the attack as well as the construction of arguments. A central notion then in the law is that of argumentation".

Thus, they consider too simplistic the idea that "once a legal text and a body of facts have been clearly represented in a formal language, the legal conclusions would follow from that representation as a matter of deduction" and for several reasons. In addition, the authors stress how, in legal reasoning, there are some activities (such as deriving legal consequences from the classified facts), where the deductive mode of reasoning has to leave "room for nonmonotonic [i.e., defeasible, our addition] techniques to deal with exceptions to rules, either statutory or based on principle and purpose, and to choose between conflicting rules on the basis of the general hierarchy of legal systems, with rules from different sources" [9,10].

Prakken and Sartor's point of view is interesting under many respects. They rely neither on the simplistic escape way of formal language (and computable law) nor on the strict rule-based decision-making, as per Frederick Schauer's view [6], and try to save the complexity of the traditional legal experience.

Further theoretical implications and crossing roads will be discussed in Section 4. Here, a final consideration is important about the very concept of legal prediction.

To predict is, in general, to state, tell about, or make known in advance. Prediction in law has several different meanings and depends on many factors, such as the object of prediction, whether an opinion of a court or a legislative or administrative decision, and depends also on the legal nature of the institutional body who delivers the decision (legislature, administration, judiciary) as well as on the legal system this body belongs to, e.g., the judiciary in a common law or in a civil system, or the time of the decision in relation to the legal, political, and cultural environment and so on.

Just to make an example, there is a significant difference based on whether the dataset used for legal analytics consists of decisions such as those by the European Court of Human Rights, which are

structured in a very uniform way, or those by the US Supreme Court, whose opinions do not have a fixed pattern and are often the result of a majority opinion of the Court, of dissenting opinions, and of concurrent opinions (i.e., opinions of judges who share the final decision with the majority, but who have reached that solution through different legal paths). Dissenting or concurrent opinions are often important because they introduce into the legal debate, also in view of subsequent decisions, legal elements, and visions that are not currently shared. For example, a concurring opinion in the Skinner case (1942 US Supreme Court) affirms, for the first time, the existence of a right to procreate as a basic right that the State cannot violate (with the legislation on eugenic sterilization), while the unanimous opinion of the Court affirmed the Habitual Criminal Sterilization Act of the Oklahoma State violated the Bill of Rights on different legal grounds [11]. Of course, even the ECHR decisions may have dissenting opinions, which, however, do not invalidate the scheme and the partitions that the Court has imposed on itself.

In a similar vein, it is as different either to decide in the very well-defined and delimited federal margin in which US Supreme Court decisions are placed, or to have to decide on 50 European jurisdictions, which also very different from one other, as it is the case for the ECHR.

The same sort of difference exists between a state-appointed or a political-appointed judge (as are the US Supreme court judges, appointed by the President with a hearing in the Senate), an appointment that legitimizes the question: "will the Justices vote based on the political preferences of the President who appointed them or form a coalition along other dimensions?"

It is clear that all the above-listed differences in points of view, theoretical approaches, and institutional aspects affect the legal, technical, or practical choices of one approach over the other in legal analytics research. Each of these features might require appropriate and different investigation tools, as we will see in the reminder of this paper.

## 3. Legal Sources and Prediction: State of the Art

In Section 2, we underlined how the very idea of prediction can encompass several operations, both from a technical and a legal viewpoint. This explains why it is quite difficult to reconstruct exactly what the state of the art of legal prediction is. Rather than focusing on technical choices and features of the most significant studies on quantitative legal prediction, we want to use them as examples of how different roads cross in the field with respect to the object of prediction and some characteristics of the used datasets. We also report the quantitative results of the studies, even if they are hardly comparable, since each study adopts a different evaluation system (and in Zhong's [3] case, the goal is also to find an automatic system of evaluation of the result, evaluation based on the comparison between summaries of decisions extracted manually and automatically).

The most significant available studies can be classified into three main categories according to their research strategies: text-based approaches, metadata-based approaches, and mixed approaches.

Text-based approaches rely only on the content of the initial data (in our case, the body of a decision) to build a structured description of the text that enables the algorithm to predict the outcome of the decision. Metadata-based approaches use background and context information, which can be inferred from the case, but do not rely directly on the text of the decision. Mixed approaches, clearly, combine the other two approaches.

The choice of one approach over the another may be based on several legal, technical, or practical reasons. For instance, from a legal standpoint, since it is an analysis conducted on decisions, the characteristics of the court and, in general, of the judicial system of origin should be significant. It is quite clear that, for example, doing legal analytics on decisions of the Italian constitutional Court, which is composed of 15 judges who serve for 9 years, is different than conducting the same kind of analysis on an ordinary Court, whose judges serve for life. With the same number of cases analyzed, the position of an individual judge in a given area will weigh more in the former case than in the latter.

From a technical perspective, decisions with metadata (entered manually and correctly) can be considered as structured data and as such allow more reliable statistical and quantitative analyses.

On the other hand, as already noticed with regard to the bottleneck concept, the manual input of this information is very time-consuming, and there is a much higher risk of human bias while entering the data.

Lastly, from a practical point of view, the choice of the approach can depend on the available data: The existence of a database containing decisions whose metadata have already been tagged, for example, can make a metadata-based approach more immediate and less time-consuming [4]; on the other hand, it is evident the availability of a dataset consisting of very structurally homogeneous decisions encourages a text-based approach. In fact, the more homogeneous the structure of the text, the more reliable the tagging can be [2].

### 3.1. Text-Based Approaches

The study published in 2016 by Aletras et al. [2] is very consistent with what we have remarked concerning the relevance in the text-based approach of the homogeneous structure of the decision. The study focuses on European Court of Human Rights decisions and has the aim to "build predictive models that can be used to unveil patterns driving judicial decisions" (p. 3).

The authors focus on the automatic analysis of cases of the European Court of Human Rights: The task is "to predict whether a particular Article of the Convention has been violated, given textual evidence extracted from a case, which comprises of specific parts pertaining to the facts, the relevant applicable law and the arguments presented by the parties involved". In order to do so, they "formulate a binary classification task where the input of classifiers is the textual content extracted from a case and the target output is the actual judgment as to whether there has been a violation of an article of the convention of human rights". They create a dataset consisting of decisions on Articles 3 (Prohibition of torture), 6 (Right to a fair trial), and 8 (Right to respect for private and family life) of the Convention. They decide to focus on these three articles for two main reasons: "First, these articles provided the most data we could automatically scrape. Second, it is of crucial importance that there should be a sufficient number of cases available, in order to test the models. Cases from the selected articles fulfilled both criteria" (p. 7).

At the beginning of the paper, they observe that their "main premise is that published judgments can be used to test the possibility of a text-based analysis for ex ante predictions of outcomes on the assumption that there is enough similarity between (at least) certain chunks of the text of published judgments and applications lodged with the Court and/or briefs submitted by parties with respect to pending cases". This supposed similarity is due to the structure of ECHR decisions, which is fixed by Rule 74 of Rules of the Court:

Rule 74: "1. A judgment as referred to in Articles 28, 42 and 44 of the Convention shall contain (a) the names of the President and the other judges constituting the Chamber or the Committee concerned, and the name of the Registrar or the Deputy Registrar; (b) the dates on which it was adopted and delivered; (c) a description of the parties; (d) the names of the Agents, advocates or advisers of the parties; (e) an account of the procedure followed; (f) the facts of the case; (g) a summary of the submissions of the parties; (h) the reasons in point of law; (i) the operative provisions; (j) the decision, if any, in respect of costs; (k) the number of judges constituting the majority; (l) where appropriate, a statement as to which text is authentic. 2. Any judge who has taken part in the consideration of the case by a Chamber or by the Grand Chamber shall be entitled to annex to the judgment either a separate opinion, concurring with or dissenting from that judgment, or a bare statement of dissent".

Assuming that semantically similar words appear in similar contexts (where by contexts they mean specific parts of the decision), the authors find the group of words that best summarizes the content (and outcome) of a decision. As they stress, "we create topics for each article by clustering together N-grams that are semantically similar by leveraging the distributional hypothesis suggesting that similar words appear in similar contexts".

The performance of the model is measured on the basis of the correctness of the prediction (since the result, the court's decision, is already known for the test cases). The average accuracy of the model

for all cases analyzed is 79%. What you can see when looking at the results is that the most important feature is constituted by "circumstances", especially when combined with information from the topics. In the case of a violation of Article 6 of the ECHR, combining these two data gives an accuracy of 84%.

An interesting study was published in 2019 by Zhong et al. [3]. The paper reports on an experiment in automatic extractive summarization of legal cases concerning post-traumatic stress disorder (PTSD) from the US Board of Veterans' Appeals (BVA). Despite the fact that, in a strict sense, this paper does text analytics for some different purposes than prediction, which is the focus of the present paper, we still refer to it because in dealing with sentences extraction, it has to perform features mapping of the text, and the techniques and reasoning developed therein are very useful for working on a text-based approach in general.

They "randomly sampled a dataset of single-issue PTSD decisions for this experiment. It comprised 112 cases where a veteran appealed a rejected initial claim for disability compensation to the BVA" (p. 5). They build a system able to summarize decision-relevant aspects of a given BVA decision by selecting sentences that are predictive of the case's outcome and they evaluate the system using a lexical overlap metric to compare the generated summaries along with expert-extracted summaries (i.e., selected sentences in the text, which summarize the decision) and expert-drafted summaries. The result of the study is evaluated (as already observed in line 178) using Rouge metrics 1 and 2, which measure the overlapping of words and word pairs. The authors compare (using Rouge 1 and 2) the results of manually extracted, drafted, and automatically extracted summaries. Comparing the work done by individual annotators (responsible for manually extracting the summaries) one obtains a very high score (greater than 0.8). The results of the comparison of the automatically extracted summaries with those extracted manually gives a result of about 0.65. Comparing the latter to sentences extracted randomly from decisions, one obtains almost identical results (around 0.63). This raises doubts about the metric chosen. The authors do not limit themselves to a quantitative analysis but also make a qualitative one, analyzing also the presence for each summary of a minimum set of information considered indispensable. This happens in about 50% of cases of automatically generated summaries.

This study, while not dealing with prediction, shows how even in this case the homogeneity of the texts was one of the reasons that led to the adoption of certain technological choices, specifically the use of text-based approaches. In this case, it is worth observing that the authors have tried to solve the bottleneck problem, both at the initial data level (they completely skipped manual initial tagging) and in the evaluation phase. In fact, they "adopt a machine learning model trained on a sufficiently large corpus, thereby bypassing the need to manually construct a factor model of the domain and train/develop factor specific language extractors/classifiers". As they state, the "research goal was to discover if outcome-predictiveness in the dataset can serve as a proxy for such domain-model-like information. After examining the results, the answer to these questions is a qualified 'no', but some insight has been gained" (p. 6).

Independently of the (self-evaluated) negative outcome of the research, it is interesting to note that even in this study the choice of the adopted technique is due to the thematic focus, structural homogeneity, and size of the dataset, which are expected to support research on automatically summarizing legal cases.

### 3.2. Metadata-Based Approach

In 2017, Daniel Martin Katz et al. [4] constructed a model designed to predict the behavior of the Supreme Court of the United States. More precisely, their purpose was to find an answer to "at least two discrete prediction questions: (1) will the Court as a whole affirm or reverse the status quo judgment and (2) will each individual Justice vote to affirm or reverse the status quo judgment?" (p. 1). In order do so, they "develop a time-evolving random forest classifier that leverages unique feature engineering to predict more than 240,000 justice votes and 28,000 cases outcomes over nearly two centuries (1816–2015)" (p. 2).

The prediction results are divided into two categories: justice level (the prediction of a single judge's vote) and case level (for the entire case decision). In the first case, the accuracy is 71.9% on data from 1816 to 2015, while in the latter case it is 70.2%. The authors (to evaluate the validity of the model) compare these results to a coin-flip (between affirm, reverse, and other), and the model again outperforms the null case by more than 50%.

It is clear Katz et al. chose a metadata-based approach for practical reasons. Indeed, the authors highlight the fact that they rely, for the work, "on data from the Supreme Court Database (SCDB). SCDB features more than two hundred years of high-quality, expertly coded data on the Court's behavior. Each case contains as many as two hundred and forty variables, including chronological variables, case background variables, justice-specific variables, and outcome variables. Many of these variables are categorical, taking on hundreds of possible values; for example, the ISSUE variable can take 384 distinct values. These SCDB variables form the basis for both features and outcome variables.

This study well exemplifies what has been observed by Ashley [1], who in his analysis focuses primarily on the use of context data for prediction, namely that "The prediction techniques make use of different types of features represented in prior cases, ranging from names of judges deciding and law firms litigating cases of different types, to attitudinal information about judges, to historical trends in decisions, to stereotypical fact patterns that strengthen a claim or defense, that is, legal factors" (p. 239). The author, however, makes a further consideration, which also seems to be applicable to the work of Katz et al.: "Such features differ in the extent to which they capture information about the merits of a case. Judges' names, law firm names, and type of case, for instance, patent litigation or product liability, capture no information about the merits of a particular legal dispute" (p. 241).

Given the availability of a large amount of data as well as their quality, it is relevant what we have already observed: It is preferable to use structured data to make quantitative analysis (and search for correlations), assuming they are correct. Obviously, the assumption is reasonable as long as the metadata is the measure of something objective and that can easily be quantified (e.g., the date of the decision, the age of the judge). Other variables cannot be easily quantified because they are intrinsically the result of a qualitative evaluation. Therefore, unlike the text-based approach, in this case, the authors work on metadata, which are already structured and rigid, and, thus, the result depends only on how they analyze the data.

## 3.3. Mixed Approaches

In a study that appeared in 2017, Nay [5] developed a machine learning approach to forecast the probability that any bill will become law in the U.S. Congress. Due to the complexity of law-making and the aleatory uncertainty in the underlying social systems, Nay chose to predict enactment probabilistically and to determine which variables are relevant in predicting enactment.

The data analyzed include all House and Senate bills from the 103rd Congress through the 113th Congress excluding simple, joint, and concurrent resolutions. For prediction, he "scored each sentence of a bill with a language model that embeds legislative vocabulary [ ... ]. This language representation enables the investigation into which words increase the probability of enactment for any topic" (p. 1).

The results of the different methodologies used for prediction are compared with the null hypothesis, i.e., that "the proportion of bills enacted in the Senate from the 103rd to the 110th Congress was 0.04 and so this is the null predicted probability of enactment of a Senate bill in the 111th Congress" (p. 7). The prediction accuracy improvement for all the methods spans from 20% to over 60%.

This study well exemplifies the mixed approach: He experiments both text-based and metadata-based approaches singularly, and then proceeds to combine them. More precisely, "to test the relative importance of text and context, [he] compared the text model to a context-only model that uses variables such as whether the bill's sponsor is in the majority party. To test the effect of changes to bills after their introduction on the ability to predict their final outcome, [he] compared using the bill text and meta-data available at the time of introduction with using the most recent data".

The conclusion is that "a model using only bill text outperforms a model using only bill context for newest data", i.e., the final version of the bill before enactment, "while context-only outperforms text-only for oldest data", i.e., the first version of the proposed bill, while "in all conditions text consistently adds predictive power".

*3.4. Some Considerations on Presented/Reported Studies*

As mentioned at the beginning of the paragraph, the scholarship surveyed here allows us to note that the choice of the method depends highly on the available data, their quality, and structure. Keeping this in mind, it is possible to make the following observations.

A text in a very homogeneous dataset with a rather rigid structure more easily allows a text-based approach. In the analysis of case law, these characteristics should be specific to each decision. This apparently neutral consideration is very problematic within a legal system. The heterogeneity of decisions is due, in fact, not only to the intrinsic heterogeneity of the subject but also to the fact that there is a plurality of courts in each system, distributed by level and matter. In addition, even among courts (distributed over the territory) that have the same competence in terms of the matter, it might be hard to impose a rigid structure, in formal or non-formal terms, on decisions. An experience in this sense will be the subject of Section 5. The attempt to impose a subsequent structure by manual tagging meets the repeatedly mentioned problem of a bottleneck.

Data homogeneity can be difficult to achieve: Firstly, because of the aforementioned heterogeneity of case law. Secondly, introducing a rigid structure (see Section 5) would create a divide with all data collected before the introduction of the new system, making it difficult to put to best use all the knowledge (i.e., previous decisions) that has been previously collected. On a more general level, therefore, the inability to modify the data available (the body of each decision) might shift the focus on the creation of a database that collects not only information that can be easily extracted manually from the body of each decision (e.g., the court that pronounced it) but also context information (e.g., the party of a bill proposer). Needlessly to say, in this case, the well-known problem of the bottleneck recurs, since not all the information (and especially that related to the context) can be extracted automatically.

The mixed approach may combine the advantages of the two other approaches (as Nay's study shows [5]) but can face the risk of duplicating the human efforts required (if we think of manual text tagging combined with manual information extraction).

## 4. Prediction of Law and Ideas about the Law

In Section 2, we gave an outline of the main roads that cross in the field of legal analytics. In Section 3, we presented the research strategies of the most significant available studies.

In this paragraph, we go back to the crossroad and take a closer look at some of the crossing roads, their feature, and some theoretical questions connected with textual legal annotation and prediction.

*4.1. Quantitative Prediction and Legal Certainty*

The old and recurrent debate on legal certainty is to a large extent connected precisely with the predictability of the decisions on disputes that arise between citizens. In this sense, legal certainty is something that goes well beyond the lawyer's, albeit reasonable, professional interest, in that it concerns the overall functioning of the system, of which the lawyer is one of the parties involved. As it has been noted: "Law is of vital importance to society, promoting justice and stability and affecting many people in important aspects of their private and public life. [ . . . ] Since law has social objectives and social effects, it must be understood by those affected by it, and its application must be explained and justified. Hence the importance of clarity of meaning and soundness of reasoning, and hence the importance of logic for the law and for legal applications of AI" [9] (p. 16).

Legal certainty has several faces: It is a principle and a value, in connection with that of equality (as in the above quoted passage); it lives in the interpretation that legal doctrine and practitioners give of it; it changes content according to the assumed idea of law, to such an extent that in a system

based on legislation as the main, if not unique, source of law (e.g., in France), it takes the meaning of certainty of legislation, while, in a system where precedents and/or the judge-made law have a greater importance, it becomes certainty of judicial interpretation (or even creation) of the law.

In common law systems, prediction is historically the prediction of judges' decisions. For instance, in US law, the prediction has been, since the end of the 19th century, identified by the famous essay by Oliver Wendell Holmes, Jr. as "the prediction of the incidence of the public force through the instrumentality of the courts" [12]. In this light, prediction is, rather than an added value to law, exactly the core of law: "a body of dogma or systematized prediction which we call the law". Thus, prediction is coessential to law, and a law that is not predictable cannot exist; it is simply not law.

In general terms, we might say that, whatever the legal organization and the cultural background, what is important is the possibility of a citizen to rely on the system and its institutions. This still holds true, even though this idea has become problematic in recent times, due to the complexity of the law-making processes and legal globalization and transnationalism [13].

Traditional (above described) legal prediction used to be the prerogative of expert lawyers and their subjective interpretation of a new case in comparison with precedents and changes in legislation. On the contrary, the new prediction ("at the center of research since 1974, with research into the outcome of tax cases") claims to be quantitative, and to be the result of the application of AI techniques.

The basic assumption is that "using a database of cases represented as sets of features and outcomes, computer programs can predict the results of new problems" [1] (p. 107, par. 4.1.).

"Machine learning techniques use feature frequency information statistically to 'learn' the correspondence between case features and target outcomes [ . . . ] A kind of ML that has been applied to predict legal outcomes is supervised ML. Since it involves inferring a classification model (or function) from labeled training data, the ML is referred to as supervised. The training data comprise a set of examples that have been assigned outcomes. Each example is a pair consisting of an input object (often a vector of feature values) and a desired output value. The learning algorithm needs to generalize from the training data to unseen situations" [1] (p. 109).

Basically, the problem of annotating the legal materials that make up the algorithm training dataset remains open: "to what extent can the features that predictive models employ be identified automatically in the case texts?" (p. 125). The road Ashley indicates is the following: "Automatically annotating case texts with features that reflect a case's legal merits, along with other argument-related information, makes it feasible to apply computational models of prediction and argumentation directly to legal texts" (p. 126). We have already seen in Section 3 an application of what is described here in theory: In particular, the description that Ashley makes helps us to understand the predictive model made by Katz et al. [4] to predict the behavior of the Supreme Court of the United States (see Section 3.2) and the summarization model of legal cases concerning PTSD from the US Board of Veterans' Appeals published in 2019 by Zhong et al. [3].

A further question involves the hidden implications and assumptions not about the dataset but about the model used. For instance, would the use of the BERT model, which is based on words and their connections with the previous and subsequent words, imply that prediction is prediction of specific connections of words? Additionally, is that specific connection of words (identified by BERT) something (semantically) meaningful per se? In the context of the analyzed decision?

We know that this is the key question in the whole field of NLP results interpretation, but here, it seems to us that the critical connection with the certainty of law is even stronger [14,15].

*4.2. Law as an Everchanging Entity: About Isomorphism*

A problem of both practical importance and theoretical interest is isomorphism in law, i.e., the condition that happens "when there is a one-to-one correspondence between the rules in a formal model and the sections of legislation modeled" [1] (p. 397). As Ashley notes:

"Since business management systems monitor compliance, the system's rules must be updated, maintained, and validated and its results must be explainable with reference to the regulatory texts.

These functions are simplified to the extent that the linkages between the logical versions of the rules and their sources in the regulatory texts are straightforward. More specifically, the legal rule modeling language needs to support isomorphism" [1] (p. 63).

However, keeping the system's rules updated and maintained, in order to have results explainable with reference to the regulatory texts as amended meanwhile, it is not easy task:

"For purposes of citing statutory texts and interweaving textual excerpts, an isomorphic mapping is essential. Isomorphic mappings between statutory text and implementing rules, however, are difficult to maintain. Frequently, the mapping is complex especially where multiple, cross-referenced provisions are involved. The versions of statutes and regulations that computers can reason with logically are different from the authoritative textual versions. Statutes may be so convoluted that even a 'faithful representation' remains unhelpful." (p. 64).

The issue is also addressed by Prakken and Sartor [9], especially in the paragraph dealing with "time and change in legal regulations" (p. 221). They recall the existing literature and, among others, refer to Governatori et al. [16] about the persistence of legal effects in a logic programming framework, using a temporal version of defeasible logic. The field is extremely important and several interesting attempts to develop defeasible logic are in progress [17].

A further response to the difficulties that isomorphism poses might overcome the present situation, where there are still two versions of the law, the first represented in natural language and the other represented in the formal model. The radical abandonment of natural language and the exclusive use of a formal language is a long-run solution, and it opens further questions that we cannot address in this paper: (i) Is it feasible? (ii) Is it conceivable that all law worldwide be represented in formal language? and (iii) Which formal language among the many (as we will see at the end of this paragraph)?

*4.3. Textual Sources, Information, Representation, and Models*

In general terms, we can say that many legal text archives are available worldwide, that they are repository of legal knowledge, and that the mission of AI techniques and quantitative legal prediction strategies is to discover this hidden legal knowledge. Another way of describing the same (or similar) reality is that AI techniques and quantitative legal prediction strategies model and structure the informational content of textual sources. A further possibility is to stress that legal texts have meanings that we present by resorting to AI techniques.

Probably all these aspects are connected or, better, they all partially overlap. Nevertheless, it is worth stressing the conceptual difference between discovering, modeling, and structuring and representing. Indeed, one thing is to discover, which means "to be the first to find, learn of, or observe" something which is supposed to be already existing or "to gain sight or knowledge of something previously unseen or unknown"; another thing is to model and structure the information, which means "to make or construct a descriptive or representational model" of something (e.g., computer programs that model climate change) and an even different thing is to represent legal meanings, which means "to indicate or communicate by signs or symbols" (meanings in quotation marks are from Farlex, the Free Dictionary).

Among the many theoretical questions these distinctions pose [18,19], and accordingly with our focus, we want only to consider a typical question in the field of legal prediction: Whether the formal representation and reasoning in law equals the human reasoning and, if not, what are the implications of such a perspective. In this connection, it is interesting to report again a passage from Ashley's work about the difference of human and machine reason and reasonableness:

"Since an ML algorithm learns rules based on statistical regularities that may surprise humans, its rules may not necessarily seem reasonable to humans. ML predictions are data-driven. Sometimes the data contain features that, for spurious reasons such as coincidence or biased selection, happen to be associated with the outcomes of cases in a particular collection. Although the machine-induced rules may lead to accurate predictions, they do not refer to human expertise and may not be as intelligible to humans as an expert's manually constructed rules. Since the rules the ML algorithm infers do not

necessarily reflect explicit legal knowledge or expertise, they may not correspond to a human expert's criteria of reasonableness." [1] (p. 111).

This is really a slippery matter. We do not dwell on in any further, and limit ourselves to noting, before leaving the field, some meaning nuances in the use of concepts, such as human legal reasoning or reasonableness, which is a well-known issue in legal linguistics [18]. In any case, we can simplify the question by saying that the ML algorithm infers something which is (at least provisionally) unknown to the world of law and legal professions. In cases like these, a first possibility is to reject from the legal domain what the ML algorithm has inferred. A second, more realistic, possibility is to accept the unforeseen content, to work on it and to make room for it in the legal field, with a final doubt: Was that inference already existing in text archives and, thus, it is only a matter of "discovery of knowledge" or is what the ML algorithm extracts something created or modeled by the same algorithm? In theoretical terms, in the second hypothesis, we should admit that if not the machine, at least the jurist who has worked on the inference, is a lawmaker or they, the human and the machine, both are.

In practical terms, i.e., thinking only whether the new entity works or not in the legal domain, this might be a less dramatic question. In the end, the AI-made law is not so different from the theoretical production of an isolated professor of law who creates a new theory about some legal argument and makes his/her theory available for lawyers, judges, and even the legislator. Depending on whether the theory is helpful in order to solve a legal problem and is in effect used, we may say that it has become law.

However, we do not discuss in this paper the delicate passages from an AI legal inference, the judicial decision and the justification that a judge gives in his/her opinions. We remain anchored to the issue of legal prediction and simply highlight how even annotation can be biased and thus play a role in the emerging of unexplainable results for humans (Ashley lists "biased selection" among the spurious reasons of such results [1]).

### 4.4. What Law Are We Talking About?

Summing up the many facets of legal prediction, we may conclude that different authors do not share and sometimes do not fully express their assumptions about what law is or should be. We think that this is relevant in order to try to make progress in understanding the mix of problems and facets that the bottleneck metaphor includes.

If we consider again Goodenough's position [8] (see Section 2), this seems to be the most radical:

"a 'computation' is any rule governed, stepwise process. These processes are surprisingly common in both the natural and human-constructed worlds [ … ] Computation theory provides a means for specifying such processes in a formal way, a bit like how the alphabet and writing allow the specifying of language formulations in print for current storage and later reconstruction. Using a computational approach, a stepwise process like a board game can be fully described in its stages, inputs and transitions. This careful specification both clarifies and records its elements. The terms of that description can be in words, in computer code, in the gears of a mechanical calculator, or even in pictures, like a flow chart. And here comes the really potent next step: a computation specified in one mode of description (including in a 'natural language' statement like the written rules for poker) can also frequently be specified in some other mode, like the binary code used by our digital computers. [ … ] Digital code and computer processing [ … ] do a lot of computational work. They have enabled us to take many of the computational processes in the world and embody them into the on/off descriptions of binary code, which our actual machines can then read and implement" [8] (p. 13).

In conclusion, "legal rules, whether set out in contracts, regulations or judicial decisions [ … ] have the same kind of stepwise branching logic [ … ]. Complexity matters, but it doesn't contradict the core point: law is often computation too" [8] (p. 16).

The key point of Goodenough's stance seems to be the adverb often. Often means "many times, frequently" and clearly differs from always. Always would have implications in the same concept of law (and its ontology), whilst often means that law may be computation and may also not, according

to its content, function, position in the system, and more. Often means that, in the present and coming technological scenario, we will have increasing parts of regulations (mostly by statutes), which can be represented, despite their complexity, directly in software: Parts but not the law as a whole and per se. Of course, the author is aware of all this and in a footnote of his 2015 paper announces, "determining the boundaries of this statement is an ongoing research project of legal informatics scholarship, including work by the author of this paper". In any case, the scenario beyond these boundaries is already conceptually clear:

"when law is computation, we can represent it in software as well. Not emulate, it in software, but represent its logic and process directly in the code. That said, in some areas of law, such as those where the processes of judicial interpretation have created not only complexity but variability and uncertainty in its specification, we can approach the modeling process from the other direction, using learning algorithms and other sophisticated data mining tools to look for emergent patterns that do emulate rather than replicate the process".

In conclusion, we might comment that, according to Goodenough, law is computation too. This happens often, and, when it happens, we can represent its logic and process directly in the code (and not emulate (i.e., imitate the function of another system) it in software. However, there are some areas of law where different approaches are needed: Learning algorithms and other sophisticated data mining tools to look for emergent patterns that do emulate the process. Thus, computation is very promising in the field of law, even though it is not able (at least till now) to cover, by definition, all the field of law and all its features.

Kevin Ashley [1] offers a multifaceted picture of the interaction between artificial intelligence tools and law, which includes also some of the fields considered by Goodenough [8], and seems to accept that the outcome of this hectic dial-up period will be multifarious. Unlike Goodenough, Ashley assumes that law is, and presumably will be, represented in natural language and that the problem that we have to face is to discover the knowledge, which is encapsulated in "text archives", a knowledge that is accessible neither at first sight nor without AI technologies.

Prakken and Sartor [9] share the focus on natural language and consider too simplistic the idea that "once a legal text and a body of facts have been clearly represented in a formal language, the legal conclusions would follow from that representation as a matter of deduction". Their criticism is based on the consideration of the many facets that the law has, beyond being a "mere conceptual or axiomatic system", and on the stress on its (of law) "orientation to future and not fully anticipated situations" the tension between the general terms of the law and the particulars of a case, and the adversarial nature of legal procedures" (p. 215). Finally, the crucial point seems to be the institutional nature of the law and the different kinds of legal norms.

## 4.5. A Step Aside in Empirical World

At this point, a clarification is needed: What law are the above-mentioned authors talking about? What are the characteristics that they assume the law has? In other words, what physics do their theories come after?

In our view, a quantitative approach to law sheds light on the sustainability of some ideas about the law of the future. According to the United Nations Environment Program the World population is currently over 7000 million people with a dramatic increase from 1500 AD when it was around 443 million people. The impact all this has over the law is evident: More people worldwide imply more contacts, more economic activity, more need for rules to be respected, and more violations and litigations. In short, we can assume the experience of law expands proportionally to the growth and needs of the human population.

The second point regards the institutional aspect of law. Nowadays, the global legal community has its unifying institution in the United Nations (UNO). Indeed, the UNO is the most encompassing worldwide legal institution. There is almost no corner in the globe that, directly or indirectly, is not covered by a UN organization.

UNO counts 193 member states, 19 intergovernmental organizations invited as observers, and 50 intergovernmental organizations. Notwithstanding the differences, each State has something that works as a legislative body, a constitution, a judiciary, and some law enforcement agencies. Thus, worldwide, there are at least 193 legislative bodies, 193 judicial systems, and 193 national police. They all, five days a week for 53 weeks a year (=265 days a year), enact laws, deliver decisions, and enforce law, and they do this using their own national natural languages, using paper and/or digital means, giving different values to those legal activities according to their specific legal systems and cultural background.

Once we take all of this into consideration, a fundamental question arises: When we talk about AI and law, which law are we talking about? Is it American law or law expressed in English? Legal multilingualism rarely occurs in the examined literature: Why? Because it is a still open and unsolved issue or because the literature is mostly written in English? What about the combination of legal (natural) multilingualism with legal (formal) multilingualism, to such an extent that a neologism has been proposed, multi<natural-formal>lingualism, in order to encompass the plurality of those combinations [20,21]? It appears that the conceptual models employed have been able to describe only a small part of the global experience of law today and only some aspects of it.

In conclusion, the global experience of law might be described as distributed in two main parts: The first is covered by computational law, something which, although quantitatively small today, is expanding its area of action (even though it does not have any conceptual and practical possibility to cover, even in the far future, the global experience of law); the second is still expressed in natural language and represents a field where there is the possibility to take advantage of the use of AI technologies and big-data analytics. This second part fragments into as many parts as are the natural languages used in the countries worldwide (where a special position has the experience of international bodies and institutions, such as the EU, where several originals written in different languages may happen to exist): This huge field is poorly explored and is waiting for the possibility offered by digital tools and big-data analytics.

*4.6. Some Consequences on Quantitative Legal Prediction*

If this is a plausible sketch of the global phenomenology of law, we might classify different situations as follows:

A. Law expressed in natural languages: This is the absolutely prevalent massive production of law materials (legislation, caselaw, administrative decisions, etc.) and includes:

   a. Rule-based decisions, i.e., decisions taken in fields of law and countries where rules exactly defined in legislation exist and, thus, the strict concept by Frederick Schauer [6] (see under Section 2) can be and is applied and expressed in one natural language.

   b. All other legal decisions/materials that do not match Schauer's standard [6] and are expressed in one specific natural language in multilingual environments, including:

      i. Different national laws worldwide.
      ii. Law materials translated from one natural language into another within international bodies and institutions.
      iii. Laws produced in several originals written in different languages within international bodies and institutions (EU, UN agencies, and more). For instance, in the EU, according to the principle of equality, language versions are not considered to be translations at all [22].

B. Computable law in its broader sense.

   a. This is the expanding area of computational contracts (as a wider category, which encompasses even smart contracts): Here, the code is law or, in other words,

law is directly expressed in code (without any passage of emulation from one system to another).

b.  Similar technologies offer regulatory paths, which are embedded in IT systems, so that the user at the end of an application procedure can directly receive a confirmation of admissibility of his/her application. Oliver Goodenough exemplifies the coming scenario: "If the United States Internal Revenue Code or its Clean Air Act were embodied in code as their original mode of enactment, a good technological parsing engine (rather than the limited biological parsing engine of a lawyer's brain) could give advice on compliance quickly and cheaply" [8] (p. 13). Examples come also from computer-assisted legal research (through LexisNexis, Westlaw, Justia.com, etc.), computer-assisted document production, computer-assisted practice management, and more. Something like this can also be applied in the judiciary, e.g., lawyers might send their briefs to a Court through the assistance of an automatic system, which at the end of the process can certify that everything is OK at least from a strict (and minimal) procedural point of view, without any need of human intervention (secretary or judicial assistants and so on). A similar system might be applied to the delivering of opinions by judges. Further examples are described in the legal literature [23,24].

c.  There are also phenomena of emulation, where pieces of legislation or contracts, originally expressed in natural language (see above A.a.), are emulated in formal languages and become computable. This happens for several aspects of legislation (even substantive criminal law), which can be expressed according to an if/then logic. This is typically the case for rule-based decisions, i.e., decisions taken in fields of law where the strict concept by Frederick Schauer [6].

d.  Decisions that are taken with a substantive contribution of machine learning systems (data-driven legal decision-making). Data might be expressed:

   i.   Directly in formal language; and
   ii.  In natural language:

      1.  And then emulated (see above under A.a); or
      2.  And not (or hardly) susceptible to be emulated.

Considering that these are some of the complex and heterogeneous ways of existence of law today, it is clear that prediction may have many different levels of application and interaction.

For instance, law in B.a., B.b., B.c., and B.d.i. might be very easily investigated and prediction might be very reliable, provided that it is properly collected and stored and that different technical tools are fully interoperable. In this case, the problems of isomorphism might disappear, assuming that changes in laws are introduced directly in a computable language and using interoperable technologies. In this (ideal) world also natural language multilingualism would disappear, and the plurality of existing formal languages is bridged adhering to a predefined rigid "data contract" that embodies the interoperability conditions.

As for A.a., legal materials should be easily emulated in computational language. As for A.b., it is the way legal materials are worldwide mostly expressed and the real working area of ML techniques for prediction. However, it is worth noting the A.b. materials (and B.d.ii.2 materials) are extremely heterogeneous for linguistic reasons (because the mere translations are not a solution for legal multilingualism), for a kind of materials, for legal background, and, last but not least, for the existence of a plurality of originals in different languages.

B.d.ii. is a field where we may have all the problems seen above (isomorphism, multilingualism, and more) and in different combinations. This is the area where the bottleneck reveals itself as an extensive crossroad.

## 5. Applicative Implications of the Relationship between the Text-Based Approach and the Structure of a Decision

In this paragraph, we investigate further the issues raised in Section 3, focusing on the text-based approach. Specifically, we take a closer look at possible practical implementations of the theoretical observations made in Section 3. To this aim, we will first of all consider an experience of legal analytics that, although it does not deal with decisions, focuses on a legal matter, such as terms of service of on-line platforms (ToS).

After that, we will focus on the Italian context: To begin with, we look at a research project currently underway on Italian case law (LAILA), and then we shift our attention to the objectives already achieved and some practical possible developments, including a tentative road map of the application of advanced AI in the judiciary.

### 5.1. Claudette and Legal Analytics for Italian Law (LAILA) Projects

In spite of the fact that it has neither case law as an object nor it aims at prediction in the strict sense of the word, the study conducted by Lippi et al. in 2018 [25] is nevertheless very interesting for our analysis. In fact, the authors: propose a machine learning-based method and tool for partially automating the detection of potentially unfair clauses (contractual provisions)" [25] (p. 117). In particular, they "offer a sentence classification system able to detect full sentences, or paragraphs containing potentially unlawful clauses. Such a tool could improve consumers' understanding of what they agree upon by accepting a contract, as well as serve consumer protection organizations and agencies, by making their work more effective and efficient, by helping them scan and monitor a large number of documents automatically" [25] (p. 118). To build their model, they use a corpus of 50 on-line consumer contracts, i.e., ToS of online platforms: "such contracts were selected among those offered by some of the major players in terms of number of users, global relevance, and time the service was established. Such contracts are usually quite detailed in content, are frequently updated to reflect changes both in the service and in the applicable law, and are often available in different versions for different jurisdictions. The mark-up was done in XML by three annotators, which jointly worked for the formulation of the annotation guidelines" [25] (p. 120).

Interestingly, in Claudette, the annotation is not limited to the identification of the unfair clauses ("in analyzing the terms of service of the selected on-line platforms, we identified eight different categories of unfair clauses, as described in Sect. 2. For each type of clause we defined a corresponding XML tag, as shown in Table"), but its main objective is to classify clauses qualitatively ("we assumed that each type of clause could be classified as either clearly fair, or potentially unfair, or clearly unfair").

However, it is reasonable that, in the near future, this type of contracts (e.g., terms of service of the selected on-line platforms) will be expressed as computable contracts. If so, it is possible to imagine that the analysis will be automatic, without the need of annotation.

In our mapping of the studies currently underway, the LAILA (Legal Analytics for Italian Law) project is especially worth mentioning. The project, which is funded by the Italian Ministry of University, involves researchers from the University of Bologna, Pavia, Torino, and Napoli, and addresses the application of methods of legal analytics (LA) to a vast and diverse set of legal information: legislation, case law, and empirical legal data. It studies the use of analytics, a mix of data science, artificial intelligence (AI), machine learning (ML), natural language processing, and statistics, in the legal domain, to extract legal knowledge, infer undiscovered relationships, and engage in data-driven predictions in the fields of patent infringement and unfair competition. At present, the research group of Pavia and Bologna are working on Italian case law legal prediction (the authors of this paper are members of the University of Pavia research group).

### 5.2. Structure of the Text and Text-Based Approach: What Kind of Materials Do We Need?

At the end of Section 3, we observed that legal texts having a high rate of homogeneity and a highly rigid structure are very suitable for a text-based approach. We also remarked that this depends,

in the case of a dataset composed of legal decisions, on how their logical consistence and their partition in blocks is dedicated to specific passages of the decision. The study on the European Court of Human Rights (ECHR) by Aletras et al. is particularly relevant: "The judgments of the Court have a distinctive structure, which makes them particularly suitable for a text-based analysis. According to Rule 74 of the Rules of the Court, a judgment contains (among other things) an account of the procedure followed on the national level, the facts of the case, a summary of the submissions of the parties, which comprise their main legal arguments, the reasons in point of law articulated by the Court and the operative provisions. Judgments are clearly divided into different sections covering these contents, which allows straightforward standardization of the text and consequently renders possible text-based analysis" [2] (p. 7).

It is worth stressing the decisions of the ECHR have this rigid structure because it is required by the rules of the court (Rule 74). The same cannot be said for other jurisdictions, such as the Italian judicial system. Nevertheless, the situation is progressively changing in Italy and some interesting steps forward having been made in recent years. In 2018, in order to make decisions more homogeneous and clear, the Supreme Council of Judiciary (*Consiglio Superiore della Magistratura*), which is the constitutional body governing the judiciary, and the National Council of Lawyers (*Consiglio Nazionale Forense*), which is the national organization of bar associations, signed a Memorandum of Understanding (*Protocollo di Intesa*) on the preliminary examination of appeals, organization of work, clarity, and conciseness in the drafting of briefs and decisions in appeal [26].

The organizations of lawyers and judges shared the idea that drafting briefs and opinions should be inspired by the principles of conciseness and clarity, with the aim of making them more functional for more efficient trials, while, at the same time, respectful of adversarial principle and of the obligation for judges to deliver clear and exhaustive opinions. In this way, decision drafting schemes and criteria for the writing briefs were drafted and proposed to judges and lawyers.

The reached agreement covers both formal and substantive profiles of opinions: Here are some examples to better illustrate the point.

I. Formal Aspects [27]:

The header of the decision must contain:

- The composition of the judging body;
- The procedural rules by which the trial was celebrated;
- The complete details of the criminal defendant;
- The specification of the defendant's position: (a) free present; (b) free absent (in absentia, limited to trials in which the institution of absentia remains and Law 67/2014 does not apply); (c) detained present or waiving with the specification whether the defendant is detained in the specific trial or for other reasons; (d) subject to a personal precautionary measure other than pre-trial detention in prison.

II. Substantive Aspects [27]:

- The reconstruction of the fact and the examination of the grounds are the essential tasks of the appeal court.
- The argumentative content of the decision must be inspired by criteria of completeness, conciseness, clarity.
- It is necessary to follow the logical order of the questions raised: preliminaries, preliminary rulings, merits. The text must be internally consistent.
- It is appropriate to argue by moving from the most decisive argument to the marginal ones. Questions common to several defendants must be dealt with before the examination of individual positions.

-       The appeal court must rule on all the requests made by the parties, even if contained in the folds of the notice of appeal, regardless of the more or less orderly formulation of the requests. Account must be taken of all the requests made at the hearing or in filed briefs.

This formal and substantive structure has several functions, boosting clarity and legal certainty in the first place. However, it can also go in the direction of facilitating the work of legal analytics on the text, without the need, as noted at the end of Section 3, to rely on a subsequent manual annotation in order to impose a structure on the text. This involves, in fact, a wasteful annotation work, at least in the first stage of algorithm training. The annotation of contextual metadata might allow a higher level of prediction to be gained.

*5.3. A Road Map for the Development of the Italian Caselaw Analytics*

Over the last 10 years, the Italian judicial system has increasingly experienced a digitalization of trials (civil, criminal, and administrative), where the same activities, previously paper based, are carried out in digital format.

According to the data published by the Italian Ministry of Justice on 30 September 2019, relating to the use of the Online Trials (*Processo Civile Telematico*, PCT, and *Processo Penale Telematico*, PPT), on a national basis, in the previous 12 months, there were 9,270,688 telematic deposits, with an increase of 97,195 compared with the previous year (significantly also the data of the telematic payments related to justice charges, fees, and taxes, reached a total of €72,260,403, according to the *Codice dell'Amministrazione Digitale*) [28].

The effects of such a digital shift have been twofold: Firstly, a reorganization of both back office and front office activities started and is in progress; and secondly, an impressive accumulation of legal materials (produced by judges and lawyers) in digital format happens. The issue at the forefront today is how to transform such an amount of data into something explorable with AI techniques and big data analytics.

The roadmap that we are proposing aims at identifying in the best possible way the steps that, in our opinion, should be taken in order to exploit the full potential of the relationship between the AI and the judiciary. This roadmap has been drawn up taking into account, as we have just seen, the technological requirements but also the organizational changes that should involve the judicial apparatus. The following are the main points:

I.      The whole technical structure of the online trials (both civil and criminal) should be updated radically, according to the technological innovations that have taken place in recent years. Just a couple of examples: (a) The problem of large file attachments should be solved in an appropriate way; (b) a material archiving system should include not only files written in natural language but also images and audio.

II.     Implement a full, efficient, and modern digitalization of all the materials of the trial. The digitalization of judicial activities is the basis for any further technological application. It must be accomplished and it must concern all the trial activities and all the parties of the trial (therefore also the lawyers, as well as the experts), as well as the registry data; it must combine the easy access for judges and users with technical characteristics suitable for data collections (data lake and data warehouse) and for advanced uses of AI; it must be modern, as it has to concern all the data and information, whatever their nature.

III.    Promoting a new digital education among judges and lawyers. Points I and II concern key technical features of the systems, but they will never be sufficient without the promotion of a digital culture among judges and lawyers. The Memorandum of Understanding signed by Supreme Council of Judiciary (*Consiglio Superiore della Magistratura)* and the National Council of Lawyers (*Consiglio Nazionale Forense)* (see above under Section 5.2) is a good starting point. It should be further developed and offered as a template by default in the software judges and lawyers use to connect to the online trials.

IV. Create data lake and data warehouse taking into account the different nature of produced materials (text files and audio/video) and the time they were produced (if before or after the *Memorandum of Understanding* template). This is strictly connected to what we stressed in Section 3: Indeed, it will be necessary to differentiate materials collected before or after the adoption of the new template. Those produced before require an intervention of cleaning and homogenization (which may also be technically laborious) but which is essential to make the mass of documents and data accumulated so far available for AI exploration. In addition, this is a situation where the bottleneck occurs, due to the fact that manual marking of decisions will be necessary at least in the training phase to establish a structure. On the contrary, the new materials will be of higher quality and therefore ready for legal analytics operations and in general for the application of AI techniques.

V. Provided all this, it is possible to start an extensive legal analytics research plan in collaboration between Ministry of Justice (*Ministero della Giustizia)*, Supreme Council of Judiciary (*Consiglio Superiore della Magistratura*), and research institutions. Only at this point, a full and useful interaction between AI and Italian case law can begin.

## 6. The Bottleneck: What Do We Know and Where Do We Go from Here

In this paper, we explored the many components and aspects that make up the bottleneck discussed by Kevin Ashely [1] and how they affect the development of quantitative legal prediction. As we know, the obstacle is that of manual annotation in the use of supervised machine learning.

We analyzed the most significant available studies on legal prediction and classified them (according to their research strategies) into text-based, metadata-based, and mixed approaches, the latter combining the former two. The final view is that the choice of one approach over the another is based on several legal, technical, or practical reasons, such as the characteristics of the court and, in general, of the judicial system or the available data: The existence of a database containing decisions whose metadata have already been tagged, for example, can make a metadata-based approach more immediate and less time-consuming [4]; on the other hand, it is evident the availability of a dataset made up of very structurally homogeneous decisions encourages a text-based approach.

We then presented and discussed some authoritative pieces of legal literature (about legal certainty and the same concept of prediction in law, isomorphism, and multilingualism, referring both to natural languages and formal languages), realizing that a step aside in the empirical world was necessary in order to clarify what law we are talking about when we discuss about law/legal prediction. The world revealed a multifarious scenario, with several different aspects, many of them having a significative relationship with legal prediction, as we described in detail.

The metaphor of a crossroad has shown a descriptive utility both for the aspects inside the bottleneck and, surprisingly, for the wider scenario.

In general terms, it seems to us that works on legal quantitative prediction are still in an early stage and that quantitative prediction is a promising and important field of research. However, in order to have an effective impact on the legal profession, the real test bench is the analysis and prediction of case law from the lower courts, in front of which most of the work of judges and lawyers takes place and which is the source of a significant amount of materials.

Finally, we tried to take advantage of the work done and we outlined a possible positive development in the AI applied to judicial activity in Italy, where the point today is how to change the considerable amount of data accumulated before lower courts in the online trials into a real experience of big data analytics.

We hope we have contributed in the common effort to open or bypass the bottleneck.

**Author Contributions:** Conceptualization, A.S.; Investigation, A.S. and G.P.; Methodology, A.S. and G.P. All authors have read and agreed to the published version of the manuscript.

**Funding:** This research received no external funding.

**Acknowledgments:** The Authors would like to thank Antonio Gelameris for his comments on the IT issues of our work.

**Conflicts of Interest:** The authors declare no conflict of interest.

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
