# Peer review of "Bottleneck or Crossroad? Problems of Legal Sources Annotation and Some Theoretical Thoughts"

_stats, doi:10.3390/stats3030024_

Round 1

Reviewer 1 Report

This article focuses on quantitative legal prediction and in particular on the bottleneck represented by the need to extract manually from legal sources the substantive legal knowledge to be used by computational models. After reviewing significant studies on quantitative legal prediction published in recent years and systematize the proposed research strategies by dividing them into three different approaches, the Authors focus on the main theoretical issues connected with textual legal annotation and prediction.  Possible developments in legal analytics applied to case law in the Italian scenario are also outlined.  

The article presents an interesting analysis of the topic. Particularly appreciable are the systematization of the approaches proposed in the relevant literature on legal prediction and the reflections on ideas about law and their implications on the applications of legal prediction. The roadmap proposed in par. 5.3. is a meaningful contribution to the development of the Italian caselaw analytics.

Points to be clarified:

  • The title refers to case law but many parts of the article refer to different types of legal source (e.g. par. 4.5 and 4.6); paragraph 3 is entitled “Case-law and prediction: state of the art”, but it discusses in more general terms the state of the art in legal prediction (and, as a matter of fact, the study presented in par. 3.3. does not refer to case law but to a (totally) different legal source). This point (focus on case law prediction or legal – including all sources of law - prediction?) should be clarified.
  • Lines 404 ff.: the issue seems to be presented in a too hasty way.
  • Which should be the aims of the research plan mentioned in line 822? It is not clear if the Authors are talking here about legal analytics or quantitative legal prediction; maybe a brief comment on the French scenario –  loi 2019-222 – (in comparison with the Italian one) should be useful.

Typos:

Line 62: a closing parenthesis is missing

Line 145: different or difference?

Line 672: the or then?

Author Response

Dear,

firstly thank you for your criticism and suggestions. Hereinafter some replies to your request of clarification:
  • "the title refers to caselaw…”: we have changed the general title of the paper into Bottleneck or crossroad? Problems of legal sources annotation and some theoretical thoughts and the title of par  3.  into Legal sources and prediction: state of the art
  • "Lines 404 ff.: the issue seems to be presented in a too hasty way”: the explanation is in par.3 where the issue is presented in a more extensive way.

  • "Which should be the aims of the research plan mentioned in line 822? It is not clear if the Authors are talking here about legal analytics or quantitative legal prediction; maybe a brief comment on the French scenario –  loi 2019-222 – (in comparison with the Italian one) should be useful”: the plan is in progress and refers primarily to legal analytics in its broader sense; a development toward quantitative legal prediction is foreseen.  
Thank you Amedeo and Giulia

Reviewer 2 Report

This paper is a review of legal analytics application to case law based on legal knowledge representation. The problem of legal knowledge acquisition bottleneck is discussed and reviewed by dividing studies on this subject into text-based approaches, metadata-based approaches and mixed approaches to legal prediction.

The paper is mainly a survey of the results in literature about judicial prediction. Different approaches are reported, however the authors are suggested to include not only an indication of the analysed methodology but also the quantitative results.
One of the characteristics of this survey is actually the lack of technical details, in favour to more legal and sociological considerations.
In this respect the paper seems not very much appropriate for the aims of the special issue it is addressed to.

An interesting part which seems worth to be extended is what is discussed in Section 5.2, about the decision drafting schemes and criteria for the writing briefs proposed to judges and lawyers. The substantive aspects here proposed, in fact, can be considered a guideline for a model-driven approach to the automatic interpretation of decisions.

Minor remark:

line 534
"(and not emulate (i.e. imitate the function of another system) it in software."
Round bracket is not closed.

Author Response

Dear,

firstly we thank you for your remarks and suggestions. Hereinafter our replies:

  • "This paper is a review of legal analytics […] however the authors are suggested to include not only an indication of the analysed methodology but also the quantitative results”: we have inserted also the quantitative results for each of the reported study. We have also inserted a warning about the impossibility of  a real comparison of those results because of the important methodological differences of the study and the differences in materials used (see lines 182-185; 252-258; 271-282; 305-310; 348-352)
  • "One of the characteristics of this survey is actually the lack of technical details, in favour to more legal and sociological considerations”: hoping we have solved the problem of lack of technical details (see above), we think that research in legal prediction is at such a preliminary level also because of the poor discussion about the relationship between technical details and legal technicalities. Our aim is to give a contribution on this way. Consequently, it has to be considered that our paper is not only a survey of the technical aspects, but also (starting from the state of the art) an investigation about the reasons of the difficulties of realising real progresses in legal prediction. Needless to say a dialogue between different disciplines is a not easy (even though of vital importance)  task.     
  • "An interesting part which seems worth to be extended is what is discussed in Section 5.2, about the decision drafting schemes and criteria for the writing briefs proposed to judges and lawyers. The substantive aspects here proposed, in fact, can be considered a guideline for a model-driven approach to the automatic interpretation of decisions”: thank you for appreciating this point. However the project is at a very preliminary stage and the elaboration of a model-driven approach is part of our current research, we hope to refer on in the next future. In addition, introducing in our paper a more detailed point on our proposal of model-driven approach would be not appropriate in a survey on published studies on legal prediction.

We are open to any further clarification might be necessary.

Very best

Amedeo and Giulia

Round 2

Reviewer 2 Report

The new version of the paper addresses the issues in my comments.